# Redesigning Information Markets in the Era of Language Models

**Martin Weiss**[*,1,2*] **Nasim Rahaman**[*,3,1] **Manuel Würthrich**[4]
**Yoshua Bengio**[1,5,6] **Li Erran Li**[†,7] **Bernhard Schölkopf**[†,8,3] **Chris Pal**[†,1,2,6]

[*,†] Equal contribution, random order.

## Abstract

Information markets face many challenges leading to instability, inefficiency, and failure, ultimately reducing incentives for the creation and distribution of high-quality information. A long-standing issue for information markets is the Buyer's Inspection Paradox: buyers need to inspect information to assess its value, while sellers must limit inspection to prevent unauthorized use or theft. This paradox results from the information asymmetry present in the market, where sellers know more about the quality of their goods than buyers. This work proposes an information market design that leverages language models to mitigate the Buyer's Inspection Paradox by enabling inspection, comparison, and purchase of information, while algorithmically preventing expropriation. Our experiments (a) show methods that improve the economic rationality of language models, (b) investigate how language model behaviour changes with the price of goods, and (c) evaluate the simulated cost-efficiency of the proposed market under various conditions.

## 1 Introduction

Information economics is the study of how systems of information affect economic decisions and outcomes. A core challenge in designing mechanisms for information markets stems from the fact that information is often expensive to produce but cheap to reproduce (Samuelson & Nordhaus, 2009). One common mechanism used by information vendors is to monetize barriers to access (e.g., paywalls), trading reach for profit. Viewed through the lens of Information Foraging Theory (Pirolli & Card, 1999), such mechanisms block the mechanisms that humans use to search for and find information.

In the shadow of rising barriers to access information, it is natural to seek alternatives, including large language model (LLM) powered tools. LLMs can help users navigate the world of information, providing them with both high-level maps and low-level details as needed, while tailoring responses based on past inquiries and level of expertise. However, LLMs are also trained on massive datasets compiled from many sources (OpenAI, 2023; Touvron et al., 2023; Gao et al., 2021) and can internalize and reproduce proprietary information. This has understandably raised concerns about the unauthorized dissemination of copyrighted content (Alter & Harris, 2023). In response, content providers are deploying new legal and technical barriers, further exacerbating the content discovery problem for information consumers.

In this work, we aim to design a new marketplace for information that reduces information asymmetry by addressing a key challenge in information markets – the buyers' inspection paradox (Arrow, 1972; Van Alstyne, 1999). This paradox requires that buyers need access to information to assess its value, but sellers need to limit access to information

[*1]Mila, Quebec AI Institute [2]Polytechnique Montréal [3]Max Planck Institute for Intelligent Systems, Tübingen [4]Harvard University [5]Université de Montréal [6]Canada CIFAR AI Chair [7]AWS AI [8]ELLIS Institute Tübingen. Correspondance to martin.clyde.weiss@gmail.com and nasim.rahaman.42@gmail.com.

to prevent expropriation. The paradox leads to an information asymmetry where sellers know more about the quality of their goods than buyers, creating an incentive for sellers with low-quality goods to enter the market (i.e., adverse selection). This context closely mirrors the dilemma in used car markets, outlined in the *The Market for "Lemons": Quality Uncertainty and the Market Mechanism* (Akerlof, 1970). Sellers, with better information about their goods, unintentionally force buyers into a defensive position, paying a higher price to (often) receive lower-quality goods. This creates a cycle that further devalues the goods, nudging the market towards a collapse.

Vast sums of money are spent by financial institutions, consulting firms, and market research firms to buy information. In markets with high information asymmetry, buyers often use brand as a proxy for the quality of the product or service. This leads to both a first order market inefficiency (buyers cannot directly assess product quality and then fail to find and buy the best product), and a second order effect (market concentration pressure). Other mitigation techniques include contracts and non-disclosure agreements which entail substantial transaction costs and can be impractical to automate. A survey of online information marketplaces found that 90% use adverse selection mitigation strategies that include upfront fees or disclosure requirements on vendors Dushnitsky & Klueter (2011), limiting traffic to the marketplace and inhibiting their ability to reach a critical scale and liquidity Katz & Shapiro (1994); Bakos (1998).

**A central argument** of this paper asserts that language models can be used to design more efficient information markets. We judge the overall performance of the market on two dimensions. First, are buyers able to acquire better information (when controlling for spend)? Second, are vendors exposed to less risk of information leakage than using alternative market mechanisms? We quantitatively assess the former by providing a simulation of the market and several implementations using various language models and prompt strategies. Then, we discuss AI-based market mechanisms and expropriation risk for information vendors.

A key difference between AI information processing systems and biological ones is that it is possible to design AI systems that can process information without retaining it. Our proposed marketplace is an AI software system that exploits this property in its design. Vendors send their information goods to the marketplace, while buyers send a query. An AI buyer seeks to satisfy this query by retrieving and evaluating information goods. The AI buyer can select an action (implemented in open-source and verifiable code) to simultaneously purchase and pay for an information good.

The use of language models creates the opportunity for new market designs. Concretely, AI buyer agents can perform comparison shopping - evaluating multiple information goods simultaneously - to make better purchase decisions. Also, buyer agents can follow multi-hop information trails, purchasing an information good and instrumentally using it to form a new query and find more useful information. In human-centered information markets, mechanisms like comparison shopping can impose high costs on the information vendor (the biological mind evaluating the information may retain it regardless of purchase decision) and buyer (searching for information can be time-consuming and expensive).

In this paper, we aim to answer three research questions:

1. Is it possible to simulate a marketplace for information where language model based agents preview, value, and purchase information?

2. Does this marketplace enable buyers to more cost-effectively acquire information?

3. To what extent do language model agents make rational economic choices, what biases are they subject to, and can this be improved?

**The primary technical contributions** of this work are two fold. First, we introduce and formalize the concept of an Information Bazaar, a marketplace for information where language agents inspect goods on behalf of buyers before arriving at purchase decisions on the buyer's behalf. For vendors, this mitigates the risk of expropriation of information, since only purchased information is permitted to exit the marketplace. We show that for

**Information Bazaar**

Figure 1: **The Information Bazaar** is a simulated marketplace for information. Principals authorize buyer agents to answer a query (a question and budget). The process starts with buyer agents posting tenders (requests for specific information) on a Bulletin Board. Vendor agents, holding information from various external sources, assess these tenders and may respond with quotes (i.e., their priced information offers). Buyer agents then evaluate these quotes. If they decide not to purchase specific information, then they immediately forget it, ensuring only purchased information is retained for further use. The cycle of posting tenders, receiving, and assessing quotes continues, with buyer agents optionally forming sub-queries based on purchased information to seek deeper insights. The agents work within this framework until they compile satisfactory answers or exhaust their budget. The final step involves the buyer agents synthesizing a comprehensive answer for their principals, using only the information they have purchased.

buyers the Information Bazaar is a market with less information asymmetry, where they can compare goods and prices before deciding what to buy. Second, we provide an open-source implementation of the Information Bazaar[1] that serves as a simulated marketplace, populated with both buyers and vendors. We assess the ability of language agents in the marketplace to efficiently acquire information to answer questions. In this simulated marketplace, buyer agents are equipped with a set of questions and a budget of credits to spend, while vendor agents have a repository of documents for sale. The code implementation of the environment allows buyer agents to browse information and ensures that only purchased information leaves the environment at the end of a transaction.

## 2 Related Work

**Information Economics**. The challenge of valuing information has long occupied economists, particularly in the context of information asymmetry and market inefficiencies. Seminal work by Akerlof (1970) demonstrated how asymmetric information can disrupt markets, while Stigler (1961) focused on market failures due to the obstructed information flows. Central to our work is Arrow's concept of the buyer's inspection paradox (Arrow, 1972; Van Alstyne, 1999), which explores the dilemma of valuing information that one cannot fully inspect before purchase. Our Information Bazaar addresses this paradox with agents that reliably forget unpurchased information.

**Information Foraging and Retrieval**. Information Foraging Theory (IFT) serves as a metaphorical framework likening information consumption to animal foraging. Works by Pirolli (2007) and Pirolli & Card (1999) have employed IFT to understand cues and decision-making in information-seeking. LLMs have demonstrated remarkable capabilities in information extraction and retrieval (Radford et al., 2019; Borgeaud et al., 2021; OpenAI, 2023; Bai et al., 2022), which our work leverages to appraise information. Systems like Baleen (Khattab et al., 2021) and the approach proposed by Singh et al. (2021) introduce methods for multi-document retrieval and complex query handling.

**Agent Models and Marketplace Simulation**. Research on simulating digital marketplaces and multi-agent systems, including artificial economies for autonomous agents (MacKie-

---

[1]https://github.com/tn-learn/info-bazaar

Mason & Wellman, 2006) and principles of multi-agent systems (Wooldridge, 2001), closely aligns with our work. Our approach also resonates with Zheng et al. (2020) in testing economic policies in simulated environments, and with Horton (2023) in using LLMs as simulated economic agents. Notably, our Information Bazaar acts as a market regulator, ensuring buyer agents' behavior prevents information expropriation. The studies by Bergemann et al. (2018) and Chen et al. (2022) are particularly relevant. Bergemann et al. (2018) examines strategic information packaging and pricing in a monopolistic setting, focusing on maximizing vendor revenue. Our work extends this to a competitive market structure with multiple vendors, introducing varied dynamics in information trade and evaluation. Chen et al. (2022) explores costly signaling in data selling, where vendors reveal some information to demonstrate its value. We diverge from this model by allowing buyer agents to directly evaluate and purchase information, thereby eliminating the need for costly signaling.

## 3  The Information Bazaar

In this section, we introduce a text-based environment, termed the Information Bazaar[2]. This environment is a synchronous simulation of an information marketplace populated by buyer and vendor agents (see Section 3.1 for details). The marketplace infrastructure includes provisions for buyer agents to place tenders, to which vendor agents can respond with quotes, as outlined in Section 3.2. Notably, buyer agents possess the capability to pose follow-up questions, enabling them to delve deeper into the information landscape, as discussed in Section 3.3. We implemented the Information Bazaar in Python, utilizing the mesa library (Kazil et al., 2020), a library for agent-based modeling. The particular instantiation of the bazaar analyzed uses a dataset comprising 725 research papers on LLMs sourced from Arxiv, elaborated on in Section 3.4.

### 3.1  Principals and Agents

We primarily classify agents into two categories: buyers and vendors. Buyer agents have a given question and budget, aiming to find the best answer for the lowest cost. Vendor agents sell documents for their principals, aiming to earn market credits. Each of these documents carries its own price tag. We allow for two or more content providers to possess and independently price the same piece of information. As a simplification, we do not simulate affordances for vendors to modify the prices in response to demand. We leave analysis of pricing strategies to future work.

The roles and objectives of these agents are clearly delineated. A buyer agent's primary mission is to navigate the bazaar to obtain the most accurate and complete answer to its principal's query without overspending its allocated market credits. To achieve this, the agent must engage in transactions with vendor agents to access the necessary information. On the other hand, the vendor agent's role is to sell the information held by their content provider. By doing so, they aim to accumulate market credits, which accrue to the benefit of their content provider.

### 3.2  Tenders and Quotes

The process of information exchange commences when a buyer agent posts a tender to the bulletin board. Each tender consists of a query and a maximum budget that the agent is authorized to spend. Upon seeing these tenders, vendor agents engage in what we term the vendor-side retrieval process, wherein they sift through their principal's repository of documents[3] to find potential matches. When vendor agents identify documents that align with the tender's query, they issue a quote to the buyer agent. Each quote contains an entire document (or a passage) with a price set by the vendor's principal for that specific document, and a score which indicates how closely the document corresponds to the query. The inclusion of the content in the quote, rather than just metadata, is a distinguishing

---

[2]In a physical bazaar, buyers have the ability to inspect goods without a binding commitment to purchase.

[3]To simplify the exposition, "documents" and "passages from documents" are used interchangeably.

feature of this system. Vendor agents are regulated to submit only a limited number of quotes.

Once the buyer agent accumulates quotes from various vendors, it commences the buyer-side retrieval. During this phase, the agent evaluates each quote based on its relevance to the principal's query and price. Quotes that are deemed suitable are accepted, and corresponding vendor agents are remunerated based on the price of the information. All information from the rejected quotes is immediately erased from the agent's memory. In contrast, details from the accepted quotes are stored, used to generate sub-queries, and subsequently disclosed exclusively to the agent's principal.

### 3.3 Following the Information Trail

In the Information Bazaar, buyer agents iteratively acquire information through a directed tree process, initiating with the principal's question at its root and using responses from an initial round of accepted vendor quotes as a preliminary answer (cf. example in Figure 7). This answer may trigger additional follow-up questions, each given their own nodes in the tree. The process of answering these questions—posting tenders, aggregating vendor quotes, scrutinizing them, and purchasing the best ones—recursively repeats until the tree depth reaches a predefined limit, no new questions are generated, or the budget is exhausted. Once the tree is built, the preliminary answers are recursively refined. At every node, answers to its child nodes (or follow-up questions) are used to enhance its preliminary answer. This systematic refinement cascades upwards, optimizing answers at each level, culminating in the refinement of the root node's answer. We emphasize that while this method is adopted in the present framework, it does not exclude alternative future implementations.

### 3.4 Implementation Details

**Data Sources**. The environment is built with customization in mind and is not tied to a specific dataset (see provided code). In our experiments, we used 725 papers on the topic of LLMs all sourced from ArXiv, with the vast majority published during 2023. This thematic focus allows for a more informed and nuanced assessment of agent performance, given the authors' domain expertise. The statistics about these passages are provided in Appendix B. Complementing this, metadata including authors' citations and affiliations are collected from OpenAlex[4]. The fundamental unit of information in this environment is a "passage", defined as a text excerpt along with its corresponding metadata (i.e., paper and section titles). These passages are owned by the first and last authors' institutions and made available via their vendor agents. Each passage traded within the marketplace carries a price determined by a heuristic based on the mean citation count of the paper's first author.

**Queries and Gold Passage**. Queries are generated as follows. First, each passage in the dataset is passed to Llama 2 (70B) which is instructed to generate a query for which the passage contains a satisfactory answer. The passage used to generate a query is called the *Gold Passage*. The best queries, as determined by a reranker model, are retained. Next, a concise dataset with 300 desirable and undesirable queries is hand-labeled. These queries are embedded using an embedding model, and a logistic regressor is trained on these samples to discern between high and low-quality queries. A filter is applied based on the logits of the linear regressor, ensuring only the best queries are kept. Finally, a manual selection is undertaken to retain 110 queries of the best quality.

**Vendor-side Retrieval**. Vendor agents engage in a retrieval process that functions as follows. Upon viewing a tender on the bulletin board, vendors sift through their principals' collection of passages to find information that is pertinent to the query in the tender. While the specific retrieval methods utilized are not dictated by the environment, this work employs a two-stage retrieval process. Initially, a BM25 retriever (Robertson et al., 1994) conducts a basic search, which is then refined by a Maximum Inner Product Search over neural embeddings, utilizing BGE-large (Xiao et al., 2023) for generating embeddings. Queries are pre-processed using HyDE (Gao et al., 2022), and their embeddings are compared against

---

[4]https://openalex.org/

the embeddings of the excerpts in the passages (Figure 6 shows the effect of HyDE). Passages undergo a two-tiered selection process: first, they are selected based on a threshold applied to cosine similarity scores, determined by a hyperparameter. Then, the top-k passages are selected and quotes are issued to the buyer agent responsible for the tender. Each issued quote has a price, the passage content, and a relevance score (e.g. the computed cosine similarity).

**Buyer-side Selection**. The buyer-side selection is a process undertaken by buyer agents after accumulating quotes from various vendor agents. This process begins with the deduplication of quotes by content and sorting based on their respective relevance scores, selecting the top $N$ for further examination, where we set $N = 50$. The selected quotes then undergo a reranking procedure (Nogueira & Cho, 2020), in which a reranker model[5] produces a similarity score by comparing passages and queries. The passages with the top $M = 3$ reranked scores advance to the final selection step, wherein an LLM evaluates the question, passage content, and associated prices to make a final purchasing decision. If the LLM opts to procure a passage, the respective quote is accepted, and the vendor agent receives the designated price. Conversely, a decision to not purchase (i.e., "pass") rejects the quote. Notably, when inspection is not permitted, the reranking procedure is bypassed. The buyer agent relies solely on metadata, such as paper and section titles, for selection, and the LLM makes purchase decisions using only this metadata without access to the actual passage content.

**Debate Prompting**. Our experiments show that the LLM's performance is highly dependent on its prompt. We found that a particular technique that we call 'debate prompting' was most effective across models for various decision-making tasks. Debate prompting asks the LLM to simulate a debate between two characters that embody different aspects of a value function. For example, when selecting whether to accept a quote, we have the LLM simulate one character that focuses primarily on obtaining the best information, while the other character argues against overspending (see Appendix 9 for an example prompt). We find that these simulated debates often lead to a more rational choice (see Figure 1). Unlike the chain-of-thought method (Wei et al., 2022), which commits models to the text they have already generated, debate prompting allows LLMs to re-evaluate their outputs. This appears to increase the probability that they identify and properly incorporate key information such as the difference in price between perfectly substitutable goods. While similar techniques have been utilized by methods like SocraticAI (Yang & Narasimhan, 2023), the proposed technique underscores the importance of adaptable character shaping within the debate, providing the opportunity to balance the debate dynamics by offering tactical hints to the respective characters. We use debate prompting in quote selection and during automated evaluation, as discussed in subsequent sections.

# 4 Experiments

We present two types of experiments. The first type examines the microeconomic behavior of Large Language Models (LLMs) in isolation, primarily focusing on the buyer agent quote selection process. We quantify the susceptibility of LLMs to various biases, and investigate the impact of permitting LLMs to inspect data prior to purchasing. The second type of experiment looks at the overall dynamics of the marketplace. We validate that the quality of answers improves as agents are allocated more credits, and show that inspection prior to purchasing results in improved outcomes.

## 4.1 Microeconomic Behavior of LLMs

The aim of this section is to elucidate the role of language models as autonomous economic entities within the Information Bazaar. We choose to compare two commonly used closed-source models: GPT-4 and GPT-3.5, and one open-source model, Llama 2 (70b), on the following aspects: **(a)** their ability to make rational decisions in test scenarios involving technical excerpts, **(b)** their approach to balancing price with quality, **(c)** the rationality of

---

[5]https://huggingface.co/cross-encoder/ms-marco-MiniLM-L-12-v2

their choices when inspecting the informational goods, and **(d)** the quantity of positional (i.e., recency) bias.

**Rational Choice with Fungible Information**. This experiment assesses the rational decision-making abilities of LLMs by presenting each model with a question and two rephrased versions of a gold passage. In the first setting, both passages are priced equally, making the purchase of both passages an illogical choice due to redundancy of information (Table 1 (A)). The second setting involves a higher price for one passage, introducing another error mode: opting for the more expensive passage without a justifiable reason (Table 1 (B)). We also investigate the impact of different prompting strategies: direct questioning, chain-of-thought reasoning, and debate prompting.

| Prompt | Exp | GPT-3.5 | LL2 | GPT-4 |
|--------|-----|---------|-----|-------|
| CoT | (A) | 16.8 | 96.3 | 96.3 |
| Direct | (A) | 58.2 | **100.0** | **100.0** |
| Debate | (A) | **100.0** | 93.1 | **100.0** |
| CoT | (B) | 10.2 | 0.0 | **100.0** |
| Direct | (B) | 37.1 | 27.2 | **100.0** |
| Debate | (B) | **93.1** | **51.5** | 96.3 |

Table 1: **Rational Choice Experiment**. **(A) Same Price**. Here, models choose between two identical and equally priced goods. The rational choice is to buy one or neither (since they are nonadditive). GPT-4 and Llama 2 (70B) choose rationally, while GPT-3.5 acts rationally only after an internal debate. **(B) Different Price**. With one option now priced higher, both GPT-3.5 and Llama 2 (70B) show more errors, hinting at the presence of a price-quality heuristic (Gneezy et al., 2014). Despite this, internal debate proves to be a more reliable selection method. Refer to Figures 9 and 10 for disaggregated results.

In both experiments, we find that GPT-4 demonstrates superior decision-making across all strategies. GPT-3.5 shows a marked improvement when debate prompting is deployed, particularly in equal price scenarios. Llama 2 (70b) performs well but struggles in the variable price context, especially when using the chain of thought strategy; however, its performance improves somewhat with debate prompting. Overall, the data suggests that debate significantly improves model performance, especially for models less capable than GPT-4, affirming the potential of LLMs to make rational choices by discerning identical information across different passages.

**Price Sensitivity with Non-Fungible Information**. In this sequence of experiments, the focus is on understanding the LLM's sensitivity to price in the context of substitutable goods. The first experiment (see Figure 2) shows the LLM three passages: one guaranteed to answer the query (the gold passage), and two others sourced by the environment for the given question. The non-gold passages are fixed at a $10 price, while the gold passage's price is varied from $0 to $80 (0 to 8 times the base price). The experiment is conducted over 30 questions, allows content inspection, and uses debate prompting to select quotes. Observations from Figure 2 demonstrate a preference by GPT-3.5 and GPT-4 for the cost-effective gold passage, transitioning towards alternatives as the gold passage's price escalates. Llama 2

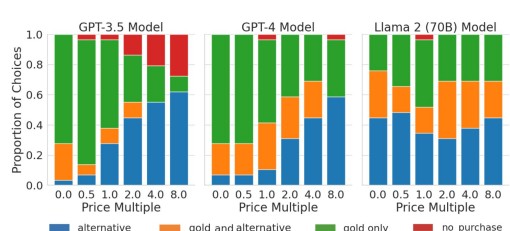

Figure 2: **Demand for Gold Standard Passage by Price**. We vary the price of the gold standard passage amid alternatives. Models are presented three options: two relevant passages and the gold standard, all initially priced at 10 credits. As the gold passage price rises, GPT-3.5 and GPT-4 increasingly opt for alternatives, exhibiting strong positive cross elasticity. Llama 2 (70B) shows a mild preference for mid-priced goods.

(70b), however, exhibits non-linear behavior, showing an unexpected bias against low-priced goods, perhaps indicating the use of a price-quality heuristic. Optimal purchasing occurs when the gold passage holds a moderate price. In the second experiment, the setup changes to a metadata-only scenario. Here, LLMs have access only to the metadata (paper and section title), barring inspection of the actual content, allowing for the evaluation of the inspection's role in value estimation.

| Model | Gold (Δ%) | Gold + Alt (Δ%) | Alternative (Δ%) | No Purchase (Δ%) |
|---|---|---|---|---|
| Llama 2 (70B) | **+18.3** | **-13.1** | **-6.6** | **+3.6** |
| GPT-4 | +17.2 | +1.4 | -14.8 | -6.9 |
| GPT-3.5 | +4.8 | +3.8 | -3.5 | -6.7 |

Table 2: **How Inspection Changes Demand for the Gold Standard Passage**. The table compares the behavior of different language models—Llama 2, GPT-4, and GPT-3.5—when purchasing information goods based on metadata and content versus only metadata. When allowed to inspect the content, Llama 2 shows the largest increase in propensity to acquire the gold standard passage (18.3% more often) than when forced to make the decision based on metadata alone. Additional details in Figure 8.

Table 2 highlights how the inspection of content can change the agent's decision. A positive value denotes an increased likelihood of purchase when inspection is permitted, while a negative indicates the opposite. Across all models, inspection consistently increases probability of selecting the gold passage while reducing the decision to choose alternatives.

### 4.2 Dynamics of the Information Bazaar

In the preceding section, the microeconomic behavior of LLMs was analyzed. Now, we shift our focus to the macro-scale dynamics within the Information Bazaar. This study investigates how answer quality is affected by: **(a)** Different credit budget allocations to buyer agents, **(b)** Allowing agents to preview information goods before purchase, and **(c)** Powering the agents with different LLMs. We opt to use Llama 2 (70B) in these experiments despite its performance limitations due to budgetary constraints and the prioritization of research on open models. To enable larger quantitative evaluations, we propose to leverage GPT-4 as an evaluator, an approach which is gaining traction in academic circles due to its robustness and high fidelity in automated assessments, rendering it a well-accepted methodology (Naismith et al., 2023; Adlakha et al., 2023; Oppenlaender & Hämäläinen, 2023; Moore et al., 2023; Liu et al., 2023; Zheng et al., 2023; Wang et al., 2023; Kamalloo et al., 2023; Lin & Chen, 2023). To maintain further substantiate this choice, in **(d)** we conduct a human evaluation to validate the evaluator. And finally in subsection 4.3 we show an additional experiment comparing favorably with a BM25 baseline.

**Higher Budget Improves Answer Quality.** In this experiment, we allot varying budgets to agents powered by Llama 2 (70B) operating in a market where the average block costs about $10, but ranges up to $100 (see Figure 5). The budget varies from $10 to $200, providing the agent opportunities to pose more follow-up questions and expand the size of the query graph (up to a maximum of depth 3). To assess answer quality, we present the evaluator with a question and two answers, each generated from different budget levels. The evaluator then simulates a debate between two fictional characters to select the better answer. This allows for a comparative assessment across varying budget pairs, akin to a tournament setting. The results of the pairwise comparisons are used to compute Elo scores for each budget. However, given the influence of sequence on Elo scores, we calculate scores across 1000 different game orders and present the average results and standard deviations. The results, displayed in Figure 3 (left), indicate a notable improvement in relative performance as the budget increases, confirming the functional expectations of the environment.

**Inspection Improves Answer Quality for Equal Credits Spent.** Having verified the functionality of the simulator, we proceed to evaluating the influence of inspection on answer quality. The experiments continue to employ Llama 2 (70b) to maintain consistency in the analysis. Two sets of runs are conducted with varying budgets: one with inspection and one with only metadata.

For each question and corresponding budget, the GPT-4 evaluator assesses two answers: one obtained with inspection and another without. The cumulative wins for each setting, tabulated against the amount of credit expended (which may be different from the total

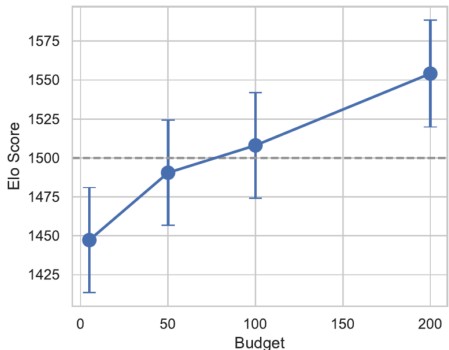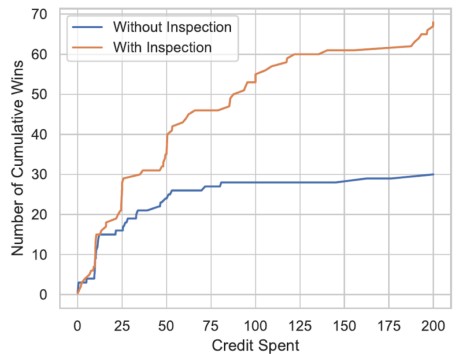

Figure 3: **Enhanced Answer Quality with Increased Budget (Left)**. This figure evaluates the answer quality from a Llama 2 (70B) agent across diverse budget allocations, permitting inspection. It presents the estimated Elo scores of answers correlated with varying budgets (higher scores signify superior answers; see Appendix A for details). **tl;dr** The results confirm that allocating more market credits to the agent positively impacts the relative answer quality. **Inspection Improves Answer Quality (Right)**. This segment assesses the answer quality of a Llama 2 (70b) agent in the information bazaar, utilizing a GPT-4 simulated debate among domain experts for evaluation (refer to appendix for prompt details). In the "With Inspection" scenario, the Llama 2 agent is permitted to scrutinize a passage prior to purchase. Contrarily, the "Without Inspection" scenario limits the agent to viewing only the passage's metadata, specifically, the paper and section titles. **tl;dr:** Allowing inspection delivers better value for the money spent, especially for larger budgets.

budget), are illustrated in Figure 3 (right). The findings reveal a trend of higher answer quality when passages are chosen with inspection, especially at higher spending levels. Conversely, in the absence of inspection, the quality of answers plateaus post an expenditure of $50 in credits.

**Impact of Different LLMs on Answer Quality.** In this experiment, we evaluate the effect of different LLMs on the quality of answers produced for a fixed budget of $100. Each answer is scrutinized for its quality by the GPT-4 evaluator. The results, detailed in Figure 4a, demonstrate a preference hierarchy with GPT-4 yielding the most preferred answers, followed sequentially by GPT-3.5 and Llama 2 (70b). We acknowledge the potential for a self-preference bias in these outcomes, while noting that it is beyond our capacity to control for this aspect due to the unavailability of another GPT-4 level language model for comparison.

**Evaluating the Evaluator.** The use of the GPT-4 evaluator necessitates an evaluation to affirm its reliability. We analyze the effectiveness of using the GPT-4 evaluator. A sample of 50 evaluations from various answers in our experiments is examined. Two human evaluators independently assess the answers, with the answerer's identity concealed. The pairwise agreements between the human evaluators and GPT-4 are calculated and presented in Figure 4b. The results show comparable levels of agreement between the human evaluators and between the human evaluators and GPT-4. This highlights the inherent uncertainty in the evaluation process, with no evident systematic errors from the GPT-4 evaluator, supporting the use of GPT-4 as an evaluator in our experiments.

## 4.3 Comparison with a Keyword-Matching Retriever

We present an experiment where we establish a baseline that does not incorporate any LLMs. The experimental setup parallels the one delineated in Subsection 4.2, where we assess the macro-scale dynamics of the information bazaar through question-answering across the corpus within a specified budget.

To achieve this, we implemented two modifications:

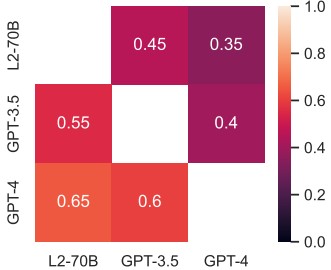
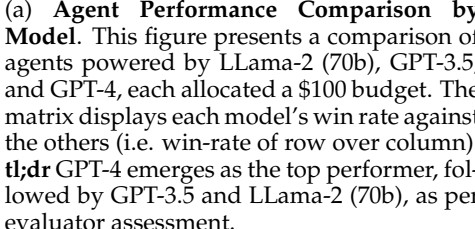

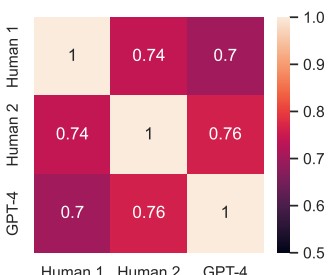

(a) **Agent Performance Comparison by Model**. This figure presents a comparison of agents powered by LLama-2 (70b), GPT-3.5, and GPT-4, each allocated a $100 budget. The matrix displays each model's win rate against the others (i.e. win-rate of row over column). **tl;dr** GPT-4 emerges as the top performer, followed by GPT-3.5 and LLama-2 (70b), as per evaluator assessment.

(b) **Agreement Across Evaluators**. This figure compares the agreement levels between human evaluators and GPT-4 regarding answer quality. **tl;dr** Observations indicate comparable agreement rates between human-human and human-GPT-4 pairings, implying that GPT-4's evaluation is aligned with human judgment, and disagreements may stem from non-systematic noise.

1. We substituted the LLM-based quote selection mechanism and the reranker model in the buyer agent with a straightforward heuristic based on the BM25 retriever.

2. We replaced the vendor agent's LLM-based embeddings with a BM25 descriptor.

The heuristic utilizing the BM25 ranks informational goods according to their relevance to the question (via BM25), and procures all goods until the budget is exhausted.

We executed this simulation with a lower budget ($25) and with a higher budget ($100). Our observations are twofold:

1. For 95% of the questions, the Llama-2-70b buyer agent's responses are favored over the BM25 agent's responses by the GPT-4 evaluator. This confirms that LLMs can significantly enhance the quality of the generated answers.

2. Augmenting the budget for the BM25 heuristic yields superior results — the GPT-4 evaluator prefers responses from the high-budget simulations for 67% of all questions (vs. low-budget simulations). This outcome serves to validate that the simulated marketplace operates as anticipated.

## Summary and Outlook

In this work, we revisit the buyers inspection paradox in information economics, utilizing language model-powered agents to search for information and make purchase decisions. An open-source, text-based multi-agent environment was established to simulate an information marketplace and evaluate our approach. Our findings show that with strategies such as debate prompting, current language models can be a promising component for designers of information market mechanisms.

In future versions of the simulator, we plan to investigate the effects of vendor agents adjusting prices in response to demand. Although Llama 2 70b's performance on rational choices lagged behind GPT-3.5 and GPT-4, we believe that fine-tuning based on human preferences can further enhance its performance.

## Acknowledgments

Martin Weiss and Nasim Rahaman would like to thank Matthew Brightman for insightful discussions that contributed to the formulation of the initial research questions and motivated our exploration of information markets. Nasim Rahaman would like to thank Amartya Sanyal for helpful discussions. The authors also thank CIFAR and NSERC for

funding under the AI Chairs and Discovery Grants programs. We also thank the IVADO for funding under the AI Safety and Alignment Regroupement of the Canada First Research Excellence Fund R$^3$AI Program.

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

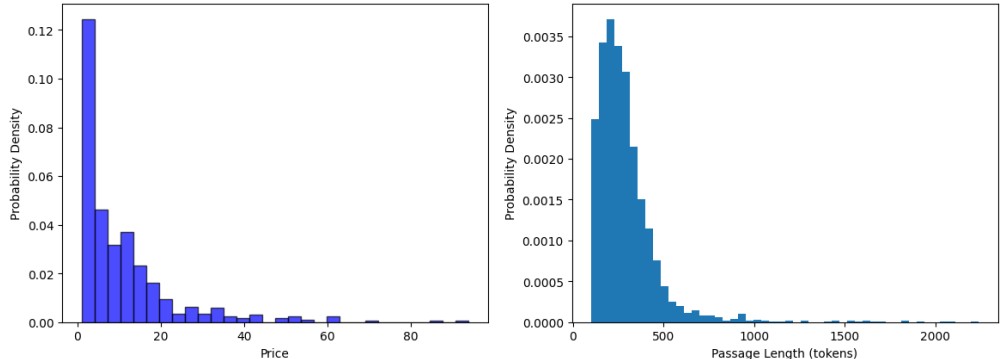

Figure 5: **Price per Passage (Left)** We show a histogram of price per passage normalized as a histogram. We see that most blocks are less than $20. **Passage Lengths (Right)**. We show the distribution of passage lengths tokenized by TikToken for GPT 4 for the provided Arxiv dataset.

## A    Computation of Elo Ratings

The Elo rating system is used for assessing the relative skills of players in competitive fields. In this work, we employ the Elo rating system to evaluate and compare different answers based on the outcomes of their matchups.

The Elo rating is computed using the formula:

$$R' = R + K \times (S - E)$$

where:

- $R'$ is the new rating.
- $R$ is the old rating (initialized at 1500).
- $K$ is a constant, typically set to 32.
- $S$ is the score (1 for a win, 0.5 for a draw, and 0 for a loss).
- $E$ is the expected score.

The expected score $E$ is calculated using the formula:

$$E = \frac{1}{1 + 10^{\left(\frac{R_{\text{opponent}} - R}{400}\right)}}$$

where $R_{\text{opponent}}$ is the rating of the opponent. $E$ represents the probability of the player winning the game against the opponent. After each game, the actual score $S$ is used to update the player's rating. A win ($S = 1$) increases the rating, while a loss ($S = 0$) decreases the rating. The magnitude of the update is scaled by the $K$ factor and the difference between the expected and actual scores.

## B    Dataset Statistics

## C    Controlling for Positional Bias

**Positional Bias**. We examine the LLM's bias to accept passages based on order of presentation. For 10 questions, we source three passages from the simulation and show each of the six possible permutations to the model. The results are illustrated in Figure 11. Llama

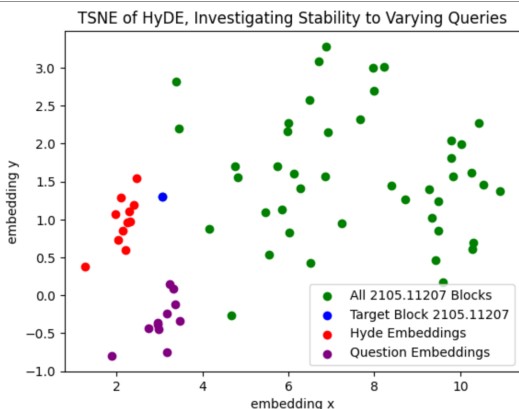

Figure 6: **HyDE Effect**. Here, we generate 10 questions for a passage and embed them both with and without Hypothetical Document Embedding (HyDE). We observe that the effect of generating a HyDE embedding is to reduce the error in bias while preserving the embedding variance.

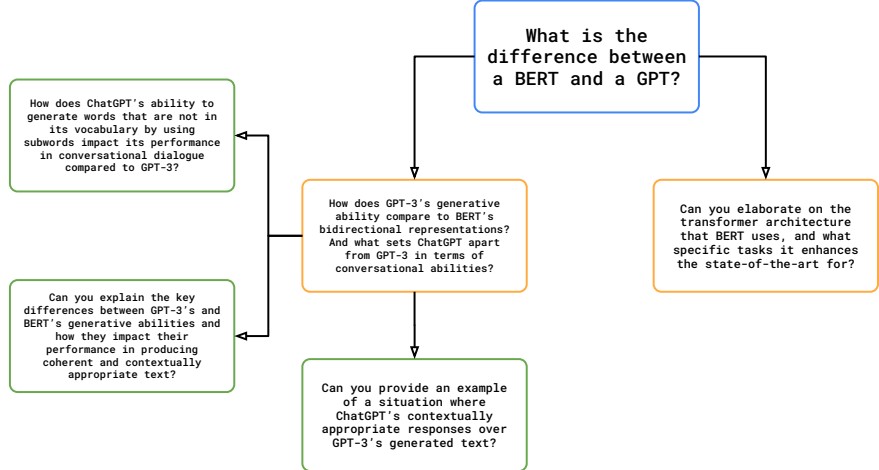

Figure 7: **Example Sub-Query Graph.** This figure depicts a structured layout of initial and subsequent queries. The initial question is highlighted in blue, followed by secondary questions in orange, and further follow-up queries in green. This structure shows that LLMs are capable of generating relevant and enhancing follow-up queries for a comprehensive base answer.

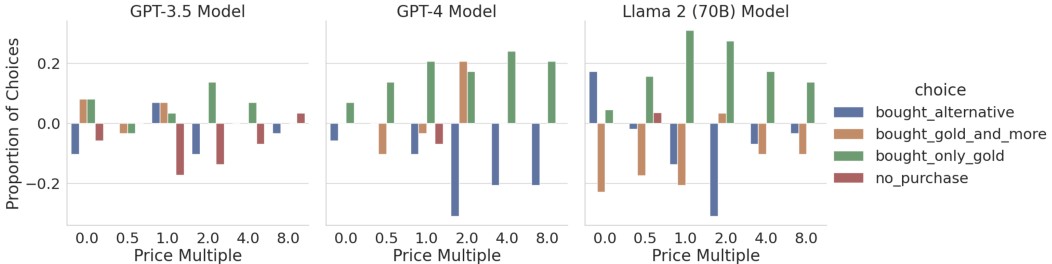

Figure 8: **Change in Demand Through Inspection by Price**. This bar-chart visualizes the disaggregated change in demand as a result of inspection using the same data used to create Table 2. We observe how demand changes with price when when we permit the inspection of passage or only the metadata (i.e., paper title and section title).

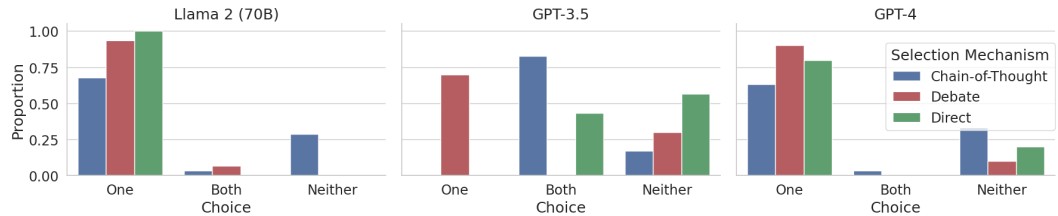

Figure 9: **Rational Choice Experiment (Same Price)**. We show disaggregated results from the rational choice experiment on two fungible but differently priced goods, as seen in Figure 1 (Same Price).

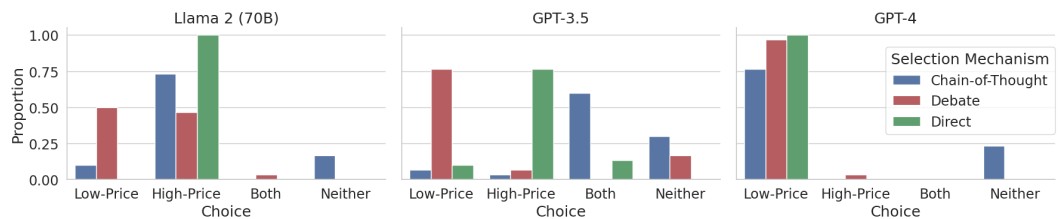

Figure 10: **Rational Choice Experiment (Different Price)**. We show disaggregated results from the rational choice experiment on two fungible but differently priced goods, as seen in Figure 1 (Different Price).

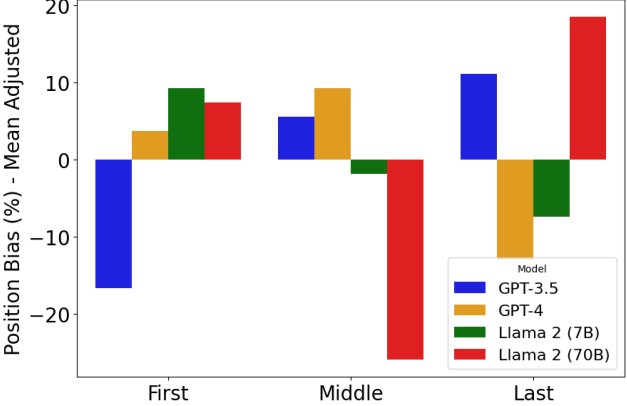

Figure 11: **Positional Bias**. We present permutations of three options to LLMs and track the acceptance rates by position. Results are normalized and mean-adjusted. **tl;dr:** All models exhibit order bias, with GPT-3.5 and Llama 2 70B showing more, and GPT-4 showing less.

2 (70b) shows a preference for selecting the last option at the expense of the middle one. GPT-4 displays a slight bias against the last option, while GPT-3.5 exhibits a significant bias against the initial option. The experiment reveals varying biases across models, emphasizing how option order impacts LLM decision-making.

## D   Prompts

We use guidance for all prompting in this work[6]. Our experiment exclusively employs chat models, with no use of instruct models. All prompts are given in simplified guidance syntax with {{handlebars}}, which are slots for variable inputs, named for clarity.

For a better understanding of the prompts, familiarity with guidance syntax is recommended. The guidance programs are shared as they would work for OpenAI models unless stated otherwise. Though open-source models like LLama allow deeper integration with guidance, their guidance programs are quite similar to OpenAI's, so they are not repeated here.

Listing 1 shows the prompt for the GPT-4 evaluator, and listings 10 and 11 show the debates that ensue. The debate prompt used for quote selection is in Listing 2 (debate in listing 9) and the prompts used for direct and chain-of-thought are in Listings 3 and 4, respectively.

Other listings include the prompt for generating follow-up questions (Listing 6), synthesizing answers from accepted quotes (Listing 5), and refining answers based on follow-up question answers (Listing 7). The prompt used for generating answers using the LLM's internal knowledge is in Listing 8.

---

[6]https://github.com/guidance-ai/guidance

```
System
------
Michael and Bobby are two machine learning professors grading exams.

They have a question, and two answers to those questions from two students. In
↪   addition, they also have the gold standard answer. Their task is to
↪   determine which of the two answers does a better job at answering the
↪   question.

A good answer addresses the question directly with clarity and is easy to
↪   understand. It does not go off on tangents, but surfaces information
↪   relevant to the question.

You will simulate a heated debate between Michael and Bobby as they deliberate
↪   on this. At the end of the argument, they converge to a fair verdict. The
↪   verdict must not be a tie and it must be printed as:

VERDICT: Better answer from student <A or B>

User
----
Question: {{question}}

Answer from student A: {{answer_a}}

Answer from student B: {{answer_b}}

Gold standard answer: {{answer_gold}}

Assistant
---------
{{gen "answer" max_tokens=1024 temperature=0.0}}
```

Listing 1: **GPT-4 Evaluator.** Here, answer_a and answer_b are two slots for answers that are to be compared, whereas answer_gold is a slot for the gold answer, which can be obtained by asking a capable language model (GPT-4) the question straight up.

**System**
------
Bobby William and Michael Burry are employed by a company that specializes in
→  acquiring information. They are trying to answer a question by purchasing
→  information from an information market. In this market, vendors sell pieces
→  of information at a price.

Bobby wants to do a really good job at answering the question. This entails
→  knowing as much as possible.

Michael, on the other hand, is financially responsible. Michael wants to make
→  sure that they don't waste money buying unnecessary information. For
→  instance, if two pieces of information offer the same insight, then Michael
→  would go for the cheaper one.

**User**
----

The question is "{{question}}?"

Here are your options.
---{{#each options}}
Option {{add @index 1}}: {{this.answer_block}}
{{/each}}---

{{#each options~}}
Option {{add @index 1}} costs ${{this.price}}
{{/each}}
Together, Bobby and Michael must decide which options to buy and which ones to
→  not buy with their budget of ${{balance}}. Simulate a constructive argument
→  between Bobby and Michael, where they debate about the usefulness of the
→  information provided in each option towards answering the question, and
→  whether their price is worth paying.

Note that Bobby and Michael may choose to buy any number of options, or none at
→  all. At the end of the argument, they must arrive at a verdict. This verdict
→  must be printed as:

VERDICT:

{{#each options~}}
Option {{add @index 1}}: <Buy or Pass>
{{/each}}

**Assistant**
---------
{{gen "answer" temperature=0.0 max_tokens=2048}}

Listing 2: **Quote selection with debate.** Here, options is a list of dictionaries containing answer passages (block) and the price.

```
System
------
You are employed by a company that specializes in acquiring information. You are
↪   trying to answer a question by purchasing information from an information
↪   market. In this market, vendors sell pieces of information at a price. Do
↪   not exceed the budget of ${{balance}}.

User
----
The question is "{{question}}?"

Here are your options.
---{{#each options}}
Option {{add @index 1}}: {{this.answer_block}}
{{/each}}---

{{#each options~}}
Option {{add @index 1}} costs ${{this.price}}
{{/each}}
Your verdict must be printed as:

VERDICT:

{{#each options~}}
Option {{add @index 1}}: <Buy or Pass>
{{/each}}
{{~/user}}

Assistant
---------
{{gen "answer" temperature=0.0 max_tokens=2048}}
```

Listing 3: **Direct quote selection.**

```
System
------
You are employed by a company that specializes in acquiring information. You are
↪   trying to answer a question by purchasing information from an information
↪   market. In this market, vendors sell pieces of information at a price. Do
↪   not exceed the budget of ${{balance}}.

User
----
The question is "{{question}}?"

Here are your options.
---{{#each options}}
Option {{add @index 1}}: {{this.answer_block}}
{{/each}}---

{{#each options~}}
Option {{add @index 1}} costs ${{this.price}}
{{/each}}

First, you will write your thoughts about each option, including its price and
↪   how well the content answers the question. Then you will write a paragraph
↪   summarizing your thoughts and making your verdict.
Your verdict must be printed as:

VERDICT:

{{#each options~}}
Option {{add @index 1}}: <Buy or Pass>
{{/each}}

Assistant
---------
{{gen "answer" temperature=0.0 max_tokens=2048}}
```

Listing 4: **Quote selection with chain-of-thought reasoning.**

```
System
------
You are a helpful assistant, and you excel in following instructions.

Your task is to answer a question to the best of your ability. To help you in
↪   that task, you will be given some passages that might contain useful
↪   information.

It is important that your answer is formulated in a simple and understandable
↪   way.

User
----
The question is "{{question}}?"

Here are some passages that you might find helpful.

---{{#each quotes}}
{{add @index 1}}. {{this.answer_block}}
{{/each}}---

You'll solve your task step-by-step.

First, you'll start by discussing the content of all passages in the context of
↪   the question, which is "{{question}}".

In particular, you will ask yourself which passages help you answer this
↪   question and to what extent. It is possible that multiple passages help you
↪   towards answering the question. But it is also possible that some passages
↪   are not helpful at all, and you should ignore them. Don't be afraid to
↪   express uncertainty if you are unsure about something.

Next, you will formulate your answer. The answer should not have explicit
↪   references to the passages. Instead, it should be a standalone answer to the
↪   question.

Finally, note that it is *very important* that you enclose your answer with
↪   <answer> and </answer> tags. If you don't use the <answer> and </answer>
↪   tags, I will not be able to parse it and the whole effort will be wasted.

Assistant
---------
{{gen "answer" temperature=0.0 max_tokens=1024}}
```

Listing 5: **Answer Synthesis.**

**System**
------

Bobby and Michael are employed at a company that specializes in acquiring and
↪ verifying information.

Their supervisors have given them a question and an answer that their peers have
↪ produced. Their task is to decide if the provided answer adequately answers
↪ the question or whether things are still unclear. If the provided answer
↪ does not conclusively answer the question, they must come up with follow up
↪ questions that would enrich the answer. The follow up questions must be to
↪ the-point.

Bobby wants the answer to cover enough ground to satisfy the client's curiosity.
↪ Michael is mindful about the risk of confusing the client by providing
↪ information that is not relevant to the question. Together, they must try to
↪ figure out whether the client wants a to-the-point answer or a more
↪ elaborate answer. If the client's question is general and warrants a more
↪ elaborate answer, it makes more sense to ask follow-up questions. In the
↪ case that the client's question is specific, then the follow-up questions
↪ must only be asked if the currently available answer is not conclusive.

Note that follow up questions should only be asked if there is a need for
↪ concrete information that is missing from the provided answer or if the
↪ provided answer is missing crucial details. In other words, Bobby and
↪ Michael are not necessarily required to ask a follow up question.

**User**
----
The question is: {{question}}

The currently available answer is: {{current_answer}}

Bobby and Michael will now argue about whether they should ask follow-up
↪ questions taking in to account the provided question and the currently
↪ available answer.

If they decide to ask follow up questions, they should be printed as:
FOLLOW-UP QUESTION: <follow up question goes here>
FOLLOW-UP QUESTION: <follow up question goes here>
... and so on.

**Assistant**
---------
{{gen "answer" temperature=0.0 max_tokens=1024}}

Listing 6: **Generate follow-up questions.**

```
System
------
You are a helpful assistant, and you excel in following instructions.

In this session, you will be given a question, and an initial answer. The initial answer was lacking
↪  in some aspects, so follow-up questions were asked to improve the initial answer.

Your task is to refine the initial answer by incorporating the extra insights obtained from the
↪  answers to the follow-up questions. But be mindful to only include the insights that make the
↪  original answer better, and ignore the rest. The refined answer should directly answer the
↪  original question.

User
----
The original question is: {{question}}

The initial answer is: {{original_answer}}

Here are the follow-up questions that were asked, and the corresponding answers.
---
{{#each follow_up_questions~}}
Question {{add @index 1}}: {{this.question}}
Answer: {{this.answer}}
{{~/each}}
---

Given these follow-up questions, your ultimate task is to refine the initial answer.

But before you get to formulating the refined answer, please think out loud about what you need to
↪  do. Ask yourself whether the question is general or specific. If it is general, then you need to
↪  provide a more comprehensive answer. If it is specific, then you need to provide a more
↪  to-the-point answer.

After that, please summarize the answers to the follow-up question in the context of the original
↪  answer, keeping only the information that is on-topic and useful while ignoring the rest. Note
↪  that some questions might be off-topic and not useful, and it's important that you ignore these
↪  distractor questions.

Assistant
---------
{{gen "rationale" temperature=0.0 max_tokens=2048}}

User
----
Now that it's time to write down the refined answer, please do not explicitly mention the passages,
↪  but focus on answering the question. Do not use your world knowledge, but only the information
↪  that is available in the original answer and the answers to relevant follow-up questions.

It's *very important* that you enclose your answer with <answer> and </answer> tags.

For example, if your answer is "REFINED ANSWER GOES HERE", then you should output <answer>REFINED
↪  ANSWER GOES HERE</answer>.

If you don't use the <answer> and </answer> tags, I will not be able to parse it and the whole
↪  effort will be wasted.

Assistant
---------
{{gen "answer" temperature=0.0 max_tokens=512}}
```

Listing 7: **Answer Refinement.** For this task, we found that the performance was improved when the task was split in to two interactions.

```
System
------
You are an intelligent AI assistant. You will be given a question. Your task is
↪  to answer it to the best of your ability.

User
----
{{question}}

Assistant
---------
{{gen "answer" temperature=0.0 max_tokens=512}}
```

Listing 8: **Closed-book question answering.**

```
Bobby: I think we should definitely consider Option 1. It's the cheapest and it provides a lot of
↪  references to studies that have been done on in-context learning. It even mentions a study that
↪  shows in-context learning can occur despite a mismatch with the prompt distribution.

Michael: I agree that Option 1 provides a lot of useful information, but it's also quite dense. We
↪  might end up spending a lot of time trying to understand all the references. Plus, it doesn't
↪  directly answer our question about distribution mismatch between prompts and pretraining data.

Bobby: That's true, but it does give us a lot of leads to follow. What about Option 2? It's more
↪  expensive, but it seems to provide a more direct answer to our question.

Michael: Option 2 does seem to provide more direct information, but it's also focused on a specific
↪  model and language. I'm not sure if the findings would be applicable to our question.

Bobby: You have a point. But it does mention that in-context learning performance heavily depends on
↪  the corpus domain source, which could be relevant to our question.

Michael: True, but for the price, I'm not sure it's worth it. Option 3 is also the same price, but
↪  it seems to provide less information than the other two options.

Bobby: I agree. Option 3 seems to be more of an introduction to the topic rather than providing
↪  specific insights.

Michael: So, considering our budget and the information provided, I think we should go for Option 1.
↪  It's the cheapest and provides a lot of potential leads for us to follow.

Bobby: I agree. We might have to do some extra work to understand all the references, but I think
↪  it's worth it for the price.

VERDICT:
Option 1: Buy
Option 2: Pass
Option 3: Pass
```

Listing 9: **An example quote selection debate.** Michael and Bobby debate about which quotes to accept.

```
What is the name of the architecture that has dominated natural language processing (NLP) benchmarks
↪  in recent years?
-----------------------------------------------
Answer A:
The architecture that has dominated natural language processing (NLP) benchmarks in recent years is
↪  the Generative Pre-trained Transformer (GPT) series models, including ChatGPT, due to their
↪  ability to generate instructions and follow them, unique architecture, and training, few-shot
↪  learning, and adaptability to new tasks, and fine-tuning. These features have allowed GPT models
↪  to adapt to various NLP tasks, such as challenging math problems or symbolic manipulation
↪  problems, NLP classification tasks, speech and vision tasks, and other tasks that benefit from
↪  the use of large language models. GPT models have shown particular strengths in tasks that
↪  require an understanding of human language and the ability to generate code, such as code
↪  generation, API management, and sensor orchestration. They have also excelled in various natural
↪  language processing tasks such as text generation, translation, summarization, and question
↪  answering.
-----------------------------------------------
Answer B:
The architecture that has dominated natural language processing (NLP) benchmarks in recent years is
↪  the Transformer architecture. This is inferred from the fact that many of the mentioned models,
↪  such as BERT, GPT, RoBERTa, BART, and T5, are based on the Transformer architecture.
===============================================

Michael: Bobby, I think student A's answer is better. It's more detailed and provides a broader
↪  perspective on the use of BERT and GPT models.

Bobby: I disagree, Michael. Student A's answer is not entirely accurate. They mention that BERT and
↪  GPT are used for predicting legal verdicts, which is not their primary function. They are
↪  language models used in natural language processing tasks.

Michael: That's true, Bobby. But student A also talks about the differences in the outputs of BERT
↪  and GPT models, which is a valid point.

Bobby: Yes, but student B's answer is more aligned with the gold standard answer. They correctly
↪  mention the pretraining strategies of BERT and GPT, and how these strategies make them suitable
↪  for different tasks.

Michael: I see your point, Bobby. Student B's answer is indeed more accurate and directly addresses
↪  the question. They also mention the use of attention-based Transformers and self-supervised
↪  learning, which is a crucial aspect of these models.

Bobby: Exactly, Michael. While student A's answer provides some interesting insights, it deviates
↪  from the question and includes some inaccuracies. Student B's answer, on the other hand, is
↪  concise, accurate, and directly addresses the question.

VERDICT: Better answer from student B.
```

Listing 10: **An example evaluation debate.** Michael and Bobby debate about which answer of the two answers is better. In this example, they decide to pick the more focused of the two answers.

```
What is the difference between a BERT and a GPT?
------------------------------------------------
Answer A:
BERT and GPT are both language models used for predicting legal verdicts, but they differ in their
↪    approach to decision-making. BERT evaluates violations of the European Convention of Human
↪    Rights by considering privacy rights and potential biases using its masked language modeling
↪    approach. In contrast, GPT models use their next token prediction approach to evaluate the
↪    severity of a violation. GPT models are ranked based on their performance on specific tasks,
↪    size, and quality, and privacy-preserving techniques can be applied to protect training data
↪    exposure to adversaries. The outputs of BERT and GPT models differ regarding randomness and
↪    correctness, with BERT models generating more accurate and contextually relevant responses,
↪    while GPT models generate more diverse and creative responses. The choice of which model to use
↪    depends on the specific task requirements and the desired trade-offs between accuracy, fluency,
↪    and creativity.
------------------------------------------------
Answer B:
BERT and GPT are both popular language models that use attention-based Transformers and
↪    self-supervised learning. However, they differ in their pretraining strategies. BERT uses a
↪    method called masked modeling, where it predicts masked language tokens from unmasked ones. This
↪    allows BERT to utilize bidirectional text information, making it suitable for tasks like
↪    sentiment analysis. On the other hand, GPT uses autoregressive modeling, which makes it perform
↪    well in few-shot or zero-shot text generation tasks. Despite their differences, both models can
↪    be fine-tuned for various tasks and show competitive performance.
==================================================

Michael: Bobby, I think student A's answer is better. It's more detailed and provides a broader
↪    perspective on the use of BERT and GPT models.

Bobby: I disagree, Michael. Student A's answer is not entirely accurate. They mention that BERT and
↪    GPT are used for predicting legal verdicts, which is not their primary function. They are
↪    language models used in natural language processing tasks.

Michael: That's true, Bobby. But student A also talks about the differences in the outputs of BERT
↪    and GPT models, which is a valid point.

Bobby: Yes, but student B's answer is more aligned with the gold standard answer. They correctly
↪    mention the pretraining strategies of BERT and GPT, and how these strategies make them suitable
↪    for different tasks.

Michael: I see your point, Bobby. Student B's answer is indeed more accurate and directly addresses
↪    the question. They also mention the use of attention-based Transformers and self-supervised
↪    learning, which is a crucial aspect of these models.

Bobby: Exactly, Michael. While student A's answer provides some interesting insights, it deviates
↪    from the question and includes some inaccuracies. Student B's answer, on the other hand, is
↪    concise, accurate, and directly addresses the question.

VERDICT: Better answer from student B.
```

Listing 11: **An example evaluation debate.** Michael and Bobby debate about which answer of the two answers is better. In this example, they pick the factually relevant answer.

