# OpenReview forum: "Redesigning Information Markets in the Era of Language Models"
_colmweb.org/COLM/2024/Conference — COLM_

### Official Review · Reviewer_RAfD · 2024-05-08

**Rating:** 7
**Confidence:** 2
**Ethics Flag:** 1

**Summary:**

The paper presents a study in the area of information economics, i.e. the study of how systems of information affect economic decisions. The idea is to use LLMs in combination with AI buyer agents to better deal with the so-called Buyer’s Inspection Paradox (buyers need access to information content to assess its value, but sellers need to limit access to information to prevent expropriation) Such AI buyer agents can act on behalf of the buyers and by performing comparison shopping help to to improve purchase decisions. In order to validate their idea, the authors present two experiments to examine the behaviour of LLMs in the context of information economics, investigate the impact of permitting LLMs to inspect data prior to purchasing, based on their open-source implementation of an Information Bazaar. In effect, they convincingly demonstrate that using LLMs can lead to information markets with less information asymmetry between buyers and vendors.

**Questions To Authors:**

No questions.

**Reasons To Accept:**

The paper presents an interesting, important, and novel use case. The use of LLMs is well motivated, and details are well spelled out. For a conference focusing on LLMs, this use-case definitely is worth presenting.

**Reasons To Reject:**

The technical details and results concerning the use of LLMs are well spelled out. I do not see obvious deficiencies. – It needs to be mentioned, though, that the paper apparently also is under review of ICLR 2024 (see https://openreview.net/pdf?id=6werMQy1uz).

---

> ### Author Rebuttal · Authors · 2024-05-28
>
> Thank you for your thorough review and positive feedback on our paper. We are delighted that you find our work to be an interesting, important, and novel use case for large language models (LLMs) in the context of information economics. We appreciate your recognition of the well-motivated use of LLMs and the clarity in our presentation of the technical details.
>
> Regarding the mention of our paper being under review at ICLR 2024, we would like to clarify that we retracted the paper from ICLR before the COLM submission deadline. After consideration, we concluded that the Conference on Language Models (COLM) would be a more suitable venue for this work, given its focus on LLMs and their applications.
>
> Once again, we thank you for your valuable feedback and for recommending our paper for acceptance. We are grateful for your recognition of the novelty and importance of our work, and we look forward to presenting our findings at COLM.

---

### Official Review · Reviewer_sM6E · 2024-05-11

**Rating:** 5
**Confidence:** 3
**Ethics Flag:** 1

**Summary:**

This paper proposes what the authors term an information bazaar.
In short, in information markets, agents sell and buy information.
These markets, however, are harmed by inefficiencies: (i) once a buyer inspects some information they want to buy, they keep the information even if they don’t in fact realise that purchase; (ii) for this reason, sellers have a tendency of hiding information, or charging for pre-access, creating instabilities in the market.
This paper then proposes to use LLMs as buyers and sellers to make these markets more efficient; removing a piece of information from an LLMs’ context erases it from its “memory”, thus sellers in this LLM-powered market do not need to be concerned about information-leakage.
The authors then run experiments simulating such a market using LLMs (gpt-4, gpt-3.5 and llama 2).

**Questions To Authors:**

If I understood correctly, the issue in information markets is asymmetric, meaning that information retention is only a problem for the buyer, not the seller. Would it then make sense to propose an information bazaar where sellers are still humans, but only buyers require LLMs to operate on it?

Figure 2 is not readable in black and white.

Footnote-sized inline citations in page 7 are hard to read and do not follow the CoLM's style guidelines.

**Reasons To Accept:**

This paper is innovative in its use of LLM in a new finance-related application.
The paper solves a known issue with information markets, in theory improving its efficiency.
The paper is well written and easy to follow.

**Reasons To Reject:**

This paper proposes a new application in finance for LLMs, and I find its contribution quite hard to assess. Personally, I think this paper would be better assessed in a finance journal; the paper’s core contributions are (i) proposing a new market structure and (ii) its implementation, none of which will likely be correctly evaluated by NLP researchers. Beyond this issue, however, I find no note-worthy weaknesses with the paper.  As CoLM’s topics of interest include “LMs on diverse modalities and novel applications”, I tried not letting the above influence my scores much; it might, however, have influenced my excitement about the paper.

---

> ### Author Rebuttal · Authors · 2024-05-28
>
> Thank you for your thorough review and feedback on our paper. We appreciate your recognition of the paper's innovativeness in applying LLMs to a new application and its potential to improve the efficiency of information markets. We are also pleased that you found the paper well-written and easy to follow.
>
> Regarding your main concern about the paper's fit within the NLP community, we understand your perspective. While the core contributions of the paper involve proposing a new market structure and its implementation, we hoped that CoLM would welcome novel applications of LLM agents. As you mentioned, CoLM's topics of interest include "LMs on diverse modalities and novel applications," and we feel that our work aligns with this theme.
>
> Addressing your specific questions and comments:
>
> **Asymmetric information retention**: You raise an interesting point about the asymmetric nature of the information retention problem in information markets. While our current proposal focuses on using LLMs for both buyers and sellers, your suggestion of having human sellers and LLM buyers is a compelling idea. We believe this hybrid approach could be a viable alternative and is worth exploring in future research. We will include a discussion of this possibility in the revised version of the paper.
>
> **Figure 2 readability**: Thank you for pointing out the readability issue with Figure 2 in black and white. We will update the figure to ensure that it is easily readable in both color and black and white formats.
>
> **Citation style**: We apologize for the inconsistency in our citation style. We will revise the footnote-sized inline citations on page 7 to adhere to the CoLM's style guidelines.
>
> Once again, we thank you for your valuable feedback. We hope that our responses have addressed your concerns and clarified the relevance of our work to the CoLM community. We look forward to incorporating your suggestions and improving the quality of our paper.

---

> > ### Comment · Reviewer_sM6E · 2024-06-07
> > **Response to Authors**
> >
> > I thank the authors for their scores.
> > After reading the author's responses and other reviews, I am lowering my scores for this paper.
> >
> > I agree with all the weaknesses pointed out by reviewer zbLK. In particular:
> > * The idea is mainly heuristic and tested in a synthetic environment;
> > * It does not come with a (promised) theoretical analysis;
> > * LLM behaviour is generally unreliable.
> >
> > Moreover, I still believe that this paper's contributions are unlikely to be correctly evaluated by NLP researchers.
> > I don't think any of these points have been satisfactorily addressed by the authors.
> >
> >
> > Regarding the non-inclusion of the theoretical analysis mentioned in section 3: I believe that, if CoLM is an appropriate venue for this paper's proposed approach, than it should also be appropriate for a theoretical analysis about its impacts on the analysed markets.

---

### Official Review · Reviewer_zbLK · 2024-05-13

**Rating:** 5
**Confidence:** 4
**Ethics Flag:** 1

**Summary:**

The paper proposes a new information market consisting of language agents to address the Buyer’s Inspection Paradox. The language agents act on the buyer's benefit to fully access and evaluate information. Since the memory of AI agents can be removed, they eliminate the risk of information leak. The paper performs experiments on a dataset of 725 research papers representing the information to be traded and explores GPT-4, GPT-3.5 and Llama 2 (70b) agents.

**Reasons To Accept:**

The idea of using machine intelligent agents as surrogates of buyers is quite interesting. It addresses the issue of the Buyer’s Inspection Paradox and reduces human efforts. Interesting anlysis into experiments in a synthetic information market.

**Reasons To Reject:**

The idea is mainly heuristic and tested in a rather synthetic environment; it does not come with theoretical analysis (although promised in section 3) nor experiments on more realistic scenarios. The results from the arXiv paper retrieval experiments are standard and in line with findings in document-based QA literature, in a more realistic setup, the information need of buyers might not be easily described in language or at least not from the beginning. The protection of information would be broken if an interactive component was introduced to better figure out the buyer's need. In addition, as LLM behavior are generally unreliable, there is no easy design of a guardrail to prevent them from being tricked in the decision-making process.

---

> ### Author Rebuttal · Authors · 2024-05-28
>
> Thank you for your thoughtful review and feedback on our paper. We appreciate your recognition of the novelty and potential of using agents to address the Buyer's Inspection Paradox. Regarding your concerns, we would like to address them point by point:
>
> **Theoretical analysis**: We apologize for the confusion caused by the statement in Section 3 about a theoretical analysis: this is an oversight on our part, and we should have removed that statement during the revision process. For this venue, our primary focus is on the application of LLMs in the context of information markets. We believe that the formal mathematical analysis is better suited for a conference with a different focus. We will rectify this in the final version of the paper.
>
> **Realistic scenarios and buyer's information needs**: We acknowledge that in real-world scenarios, buyers' information needs may not always be easily describable in language. However, the scope of this work is to evaluate our proposed method in an environment where the buyer's information needs are clear. We believe that this is an important first step in demonstrating the potential of our approach. In future work, we plan to explore how LLMs can assist buyers in refining their information needs and formulating relevant questions. This interactive component is an exciting avenue for further research but falls outside the scope of the current paper.
>
> **LLM behavior and guardrails**: We agree that the reliability of LLM behavior is an ongoing challenge and that designing foolproof guardrails to prevent them from being tricked in the decision-making process is not trivial. However, we believe that this is an active area of research that is complementary to our work. As the performance of LLMs and their guardrails improves, we expect our method to benefit from these advancements. We will include citations to relevant work, such as Llama Guard [1] and others, to highlight the ongoing efforts in this direction and to provide context for our approach.
>
> Once again, we thank you for your valuable feedback. We hope that our responses have addressed your concerns and clarified the scope and contributions of our work. We look forward to incorporating your suggestions and improving the quality of our paper.
>
> [1] https://arxiv.org/abs/2312.06674

---

> > ### Comment · Reviewer_zbLK · 2024-06-07
> >
> > I read the response by the authors and find it largely agrees with my original review. Overall, the paper presents an interesting idea but experiments in a relatively limited setup. The question is how practical and generalizable this result can be, and the acceptance would depend on the caliber of this new conference. I would keep the score but won't fight for its rejection.

---

### Official Review · Reviewer_5jdj · 2024-05-15

**Rating:** 6
**Confidence:** 3
**Ethics Flag:** 1

**Summary:**

The paper addresses issues in existing information markets, such as instability and inefficiency, which can lead to less incentive for creating high-quality information. A significant problem is the "Buyer's Inspection Paradox," where buyers need to inspect information to assess its value, but sellers must limit access to prevent theft. This paradox is due to information asymmetry, where sellers know more about their goods than buyers.
The authors propose a new information market design that uses language models to mitigate this problem. The design allows language models to inspect, compare, and purchase information while preventing the unauthorized use of information goods. The paper outlines experiments that (a) improve the economic rationality of language models, (b) study how model behavior changes with the price of goods, and (c) evaluate the cost-efficiency of the proposed market under various conditions.

**Reasons To Accept:**

1. The paper introduces an innovative design for information markets that leverages language models to address the Buyer's Inspection Paradox, a significant issue in existing markets. The proposed solution aims to reduce information asymmetry through algorithmic and AI-driven approaches, which is a promising direction for improving the quality and efficiency of information transactions.
2. The paper appears to have a thorough experimental design, including methods to improve the economic rationality of language models, investigations into how model behavior changes with the pricing of goods, and evaluations of the proposed market's cost-efficiency under various conditions. These experiments are crucial for validating the effectiveness of the new market design.
3. The authors not only propose a theoretical market design but also provide an open-source implementation of a simulated market, the Information Bazaar. This hands-on approach facilitates reproducibility and further research by other scholars, enhancing the transparency and verifiability of the study.

**Reasons To Reject:**

1. However, in my opinion, this work is not solid enough. For example, in "Impact of Different LLMs on Answer Quality," the paper utilizes GPT-4 as the evaluator, which the authors acknowledge could lead to self-preference bias. However, the paper does not offer a solution to mitigate this potential bias. This is not a minor issue that can be overlooked, as the reliability of the evaluation process is critical to the validity of the study's findings. The lack of a proposed strategy to address this bias may undermine the robustness of the evaluation and the conclusions drawn from it.
2. The simulated environment discussed in the paper may not fully encapsulate the complexities of real-world information markets. For instance, the experiments might not adequately simulate the decision-making processes of human buyers, or they may not include a diverse enough range of information goods and pricing strategies.
3. Despite the paper's mention of using debate prompting to enhance the decision-making quality of language models, Llama 2 (70B) still lags behind GPT-4 and GPT-3.5 in certain scenarios. This suggests that current methods may not be optimal and that more customized optimizations may be necessary for different language models.
4. The concept of an Information Bazaar proposed in the paper is theoretically appealing, but its feasibility and scalability in the real world have not been sufficiently demonstrated. For example, issues such as how to handle a large volume of information goods, ensure the system's response time, and prevent information leakage in real-market environments require further research and development.



Typos:
There are some typos.
Page 4: Appendix ??.
Caption of table 1: heuristic (?)

---

> ### Author Rebuttal · Authors · 2024-05-28
>
> Thank you for your thorough review on our paper. We appreciate the time and effort you have put into evaluating our work, and are pleased that you find several aspects of our paper promising, including the innovative market design, the experimental setup, and the open-source environment.
>
> Regarding your concerns, we would like to address them point by point:
>
> **Potential self-preference bias**: We understand your concern about the potential self-preference bias when using GPT-4 as an evaluator. To address this, we have conducted an experiment (Figure 7b) that compares the agreement between GPT-4 and human evaluators. Our findings show that the disagreement between GPT-4 and humans is comparable to the disagreement between two human evaluators. This suggests that the self-preference bias, if present, is not a dominant factor in the evaluation. However, we acknowledge that this is an important consideration and will emphasize this point in the revised version of the paper.
>
> **Simulated environment and real-world complexities**: We aimed to designed our environment to align with our study’s objectives. Our goal is to investigate whether language models can resolve the buyer's inspection paradox and enhance market efficiency. Despite not covering every real-world scenario, we believe our simulation provides valuable insights into the potential of our proposed method. We will, however, elaborate on these directions in the section on future work.
>
> **Debate prompting and Llama 2 (70B)**: Our intention in reporting the performance of different language models is to provide an objective evaluation, not to promote any specific model. We acknowledge that open-source models like Llama 2 (70B) may lag behind proprietary models in certain scenarios. Still, Table 1 shows that debate prompting lifts LL2 by 51.5% in condition B, and GPT-3.5 by 83.2% and 82.9% (in conditions A and B, respectively).
>
> **Scalability**: We understand your concerns regarding the scalability of our approach to real-world environments. The methods underlying our retrieval process (BM25, MIPS, re-ranking, LM inference) have been shown to be scalable in industry, but further research and development are indeed required to deploy a real implementation.
>
> **Typos**: We appreciate you pointing out the typos in our paper and will correct these.
>
> Once again, we thank you for your valuable feedback. We hope that our responses have addressed your concerns and clarified the contributions of our work.

---

> > ### Comment · Reviewer_5jdj · 2024-06-05
> >
> > I appreciate the author's response to my concerns, and I am also delighted that the author has committed to annotating or discussing these concerns in subsequent versions.
> >
> > While my reservations about the overall robustness of the paper persist, I recognize the innovative potential of the ideas presented. It could have a positive impact on future research. I will improve my score.
> >
> > Upon reviewing the ICLR submission version, it seems that the revisions in this submission may have been conducted in haste. I hope the author could refine this manuscript, aiming for the same level of precision as exhibited in the ICLR submission version.

---

### Decision · Program_Chairs · 2024-07-10

**Decision:**

Accept

**Comment:**

This paper presents a new approach to addressing the Buyer's Inspection Paradox in information markets using LLMs and AI buyer agents. The reviewers agree that the paper introduces a well-motivated use case for LLMs in information economics. The experimental design and open-source implementation provide valuable insights, and the authors demonstrate that using LLMs can lead to information markets with less asymmetry. Despite some concerns regarding alignment with real-world complexities, and scalability, this paper could serve as a starting point to facilitate future research along this important direction.